# INDISCRIMINATE POISONING ATTACKS ARE SHORTCUTS

## ABSTRACT

Indiscriminate data poisoning attacks, which add imperceptible perturbations to training data to maximize the test error[1], have become a trendy topic because they are thought to be capable of preventing unauthorized use of data. In this work, we investigate why these attacks work in principle. We find that the perturbations of advanced attacks are almost **linear separable** when assigned with the target labels of the corresponding samples. This is an important population property for various perturbations that were not unveiled before. Moreover, we further confirm that linear separability is indeed the workhorse for recent attacks. We synthesize linear separable data as perturbations and show such synthetic perturbations are as powerful as the deliberately crafted attacks. Our finding also suggests that the *shortcut learning* problem is more serious than previously believed as deep models heavily relies on shortcuts even if they are of an imperceptible scale and mixed together with the normal features. It also suggests that pre-trained feature extractors can be a powerful defense.

## 1 INTRODUCTION

Big datasets crawled from the Internet keep advancing the state-of-the-art results (Devlin et al., 2018; He et al., 2020; Chen et al., 2020). However, there are increasing concerns about the unauthorized use of personal data (Hill & Krolik, 2019; Prabhu & Birhane, 2020; Carlini et al., 2020). For instance, a private company has collected more than three billion face images to build commercial face recognition models without acquiring any user consents (Hill, 2020). To address those concerns, many

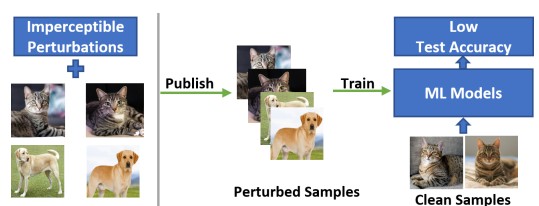

Figure 1: An illustration of indiscriminate poisoning attacks.

data poisoning attacks have been proposed to prevent data from being learned by unauthorized deep models (Feng et al., 2019; Shen et al., 2019; Huang et al., 2021; Yuan & Wu, 2021; Fowl et al., 2021a;b). They add imperceptible perturbations to the training data so that the model accuracy on unseen data is arbitrarily bad. We refer to these attacks as *indiscriminate poisoning attacks* as the adversary targets at **all** test examples. This type of attack is also know as *availability attack* (Biggio & Roli, 2018) or *delusive attack* (Tao et al., 2021). We note that the word is also used to denote the adversary does not have a specific target class (Muñoz-González et al., 2017). In Figure 1, we give an illustration of the attacks studied in this paper.

Roughly speaking, there are three methods available to construct the indiscriminate poisoning attack[2]. The first method formulates the perturbations as the solution of a bi-level optimization problem (Biggio et al., 2012; Feng et al., 2019; Fowl et al., 2021a; Yuan & Wu, 2021). The bi-level optimization problem requires models trained on perturbed data to have the maximum loss on unseen

---

[1]There are some attacks have the same objective but they inject malicious training samples instead of perturbing existing ones, e.g., Biggio et al (2012). In this work, we focus on the latter approach.

[2]This paper focuses on recent attacks that are designed for deep neural networks. Some earlier attacks against SVM are not covered in this work.

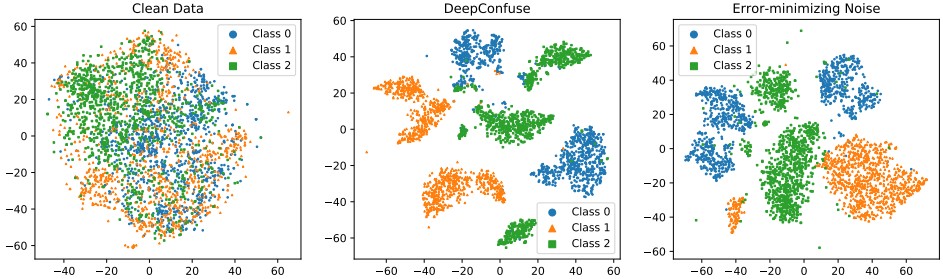

Figure 2: T-SNEs of the first three classes of clean CIFAR-10 data and the perturbations generated via DeepConfuse (Feng et al., 2019) and error-minimizing noises (Huang et al., 2021). The perturbations are flattened and normalized into unit norms.

data. Then Huang et al. (2021) conceive a simpler poisoning attack called *error-minimizing noises*, where the perturbation on training data is crafted by minimizing the training loss so that nothing is left to backpropagate in the regular training procedure. The last method uses common adversarial examples as an indiscriminate poisoning attack (Nakkiran, 2019; Tao et al., 2021; Fowl et al., 2021b). Despite of these quite different approaches, they all give strong poisoning attacks. Intrigued by this observation, we ask the following question:

*What is the underlying workhorse of recent indiscriminate poisoning attacks?*

To answer the above question, we first take a closer look at the perturbations of existing attacks. We visualize the perturbations of two advanced attacks via two-dimensional T-SNEs (Van der Maaten & Hinton, 2008) in Figure 2. Surprisingly, the perturbations with the same class label are well clustered, suggesting that the perturbations would be **linear separable** in the original high-dimensional space. We confirm this by fitting the perturbations with linear models. The perturbations are assigned with the labels of their target examples. It turns out that simple softmax regression models can fit the perturbations of four representative attacks with $> 90\%$ accuracy. This finding suggests that using linear separable perturbations may be the key for an indiscriminate poisoning attack to succeed.

To further confirm that the linear separability is a sufficient (not only necessary) condition, we reverse the above procedure: synthesizing some linear separable data to see if they can serve as data poisoning attacks. Extensive experiments on benchmark datasets and models demonstrate that the synthetic perturbations can be as powerful as advanced indiscriminate poisoning attacks. Notably, generating synthetic data as perturbations is significantly easier and cheaper than existing attacks as it does not require solving any optimization problems. This finding reveals that linear separability is indeed the workhorse to the success of recent indiscriminate poisoning attacks.

The above finding coincidentally matches a recent concept called *shortcut learning* (Geirhos et al., 2020). Shortcut learning stands for the behavior that deep models tend to rely on features that do not generalize on realistic test data. Such features are referred to as shortcuts. With this concept, the perturbations of indiscriminate poisoning attacks are also shortcuts. We note that the shortcuts in previous works are usually part of natural data, which are somehow heuristic, e.g., "grass" is a shortcut for recognizing "cow" in natural images (Beery et al., 2018; Geirhos et al., 2020). In this work, we expose a more explicit form of shortcuts and discuss extensively how to construct such shortcuts. We unveil that deep learning models would overwhelmingly rely on spurious shortcuts even though the shortcuts are scaled down to an imperceptible magnitude. This finding exposes a fundamental vulnerability of deep models and hence may be of independent interest to the community.

Finally, our understanding of the working principle of indiscriminate poisoning attacks also motivates a powerful defense against them. As the attacks succeed by providing shortcuts, they will be less effective if those shortcuts are filtered out. To achieve this, we use pre-trained models as feature extractors. If a pre-trained model is trained on datasets that have similar distributions as the target dataset, it can extract useful features and avoid shortcuts. Our experiments suggest that training only the linear classification layers with the features extracted from the pre-trained model can successfully defend existing attacks.

Our findings and contributions are summarized as follows:

- We find that the perturbations used in several advanced indiscriminate poisoning attacks are (almost) linear separable. We further validate that this property is the workhorse for the perturbations to be effective. To the best of our knowledge, we are the first to unveil this important property, which is fundamental to understand existing poisoning attacks and may inspire future attacks.

- We connect this property to the shortcut learning problem and demonstrate the omnipresence of shortcuts in learning. We show that one can construct invisible shortcuts that machine learning models would heavily rely on. We believe this finding greatly widens the understanding of shortcuts in machine learning.

- Motivated by breaking the shortcuts, we use pre-trained feature extractors to defend against indiscriminate poisoning attacks. Experiments show the proposed defense is very powerful.

## 1.1 RELATED WORK

**Data poisoning.** In general, data poisoning attacks perturb training data to intentionally cause some malfunctions of the target model (Biggio & Roli, 2018; Goldblum et al., 2020; Schwarzschild et al., 2021). A common class of poisoning attacks aims to cause test-time error on some given samples (Koh & Liang, 2017; Muñoz-González et al., 2017; Chen et al., 2017; Koh et al., 2018; Shafahi et al., 2018; Zhu et al., 2019; Shan et al., 2020; Geiping et al., 2020; Huang et al., 2020; Cherepanova et al., 2021) or on all unseen samples (Biggio et al., 2012; Feng et al., 2019; Liu & Shroff, 2019; Shen et al., 2019; Huang et al., 2021; Yuan & Wu, 2021; Fowl et al., 2021a;b). The latter attacks are also known as indiscriminate poisoning attacks as they do not have specific target examples (Barreno et al., 2010). Backdoor attack is another type of poisoning attack that perturbs training data so that the attacker can manipulate the target model's output with a designed trigger (Chen et al., 2017; Shafahi et al., 2018; Turner et al., 2018; Xie et al., 2019; Bagdasaryan et al., 2020; Nguyen & Tran, 2020; Saha et al., 2020; Nguyen & Tran, 2021). In this work, we investigate indiscriminate poisoning attacks and reveal the workhorse of them. We show that the perturbations of advanced attacks are (almost) linear separable. We further confirm that using linear separable perturbations is a sufficient condition to perform strong attacks.

There are defenses against indiscriminate poisoning attacks. Huang et al. (2021) show training with advanced data augmentation methods improves the test performance. Another defense is to train the target model with *adversarial training* (Madry et al., 2018), which currently achieves the best performance (Huang et al., 2021; Fowl et al., 2021b; Tao et al., 2021). Recently, Radiya-Dixit & Tramèr (2021) challenge the security of poisoning attacks from another view. They argue that unless existing attacks can fool all future defenses, they cannot protect the data well because the adversary can simply save the perturbed data and wait for better defenses[3]. In this work, we propose to use pre-trained models to defend indiscriminate poisoning attacks. Our experiments demonstrate that using pre-trained models is a powerful defense against indiscriminate attacks. We note that Cinà et al. (2021) have explored running poisoning attacks against pre-trained feature extractors. Nonetheless, they focus on backdoor attacks and do not advocate using pre-trained models as a defense.

**Shortcut learning.** Recently, the community has realized that deep models may rely on shortcuts to make decisions (Beery et al., 2018; Niven & Kao, 2019; Ilyas et al., 2019; Geirhos et al., 2020; Huh et al., 2021). Shortcuts are spurious features that are correlated with training labels but do not generalize on test data. Beery et al. (2018) show that a deep model would fail to recognize cows when the grass background is removed, suggesting that the model takes "grass" as a shortcut for "cow". Niven & Kao (2019) show that large language models use the strong correlation between some simple words and labels to make decisions, instead of trying to understand the sentence. For instance, the word "not" is directly used to predict negative labels. In this work, we show the shortcut learning problem is more serious than previously believed. Our experiments in Section 3 demonstrate that deep models only pick shortcuts even if the shortcuts are scaled down to an imperceptible magnitude and mixed together with normal features. These experiments reveal another form of shortcut learning, which has already been unconsciously exploited by indiscriminate data poisoning attacks. We note that there also exist other synthesize datasets that offer a stratification of features (Ross et al., 2017;

---

[3]Radiya-Dixit & Tramèr (2021) only verify this is achievable for targeted poisoning attacks (Shan et al., 2020; Cherepanova et al., 2021) though the general idea may also apply to indiscriminate poisoning attacks.

Jacobsen et al., 2018; Hermann & Lampinen, 2020; Shah et al., 2020). Those synthetic data contain shortcuts that can not be used as perturbations as they are visible and affect the normal data utility, e.g., vertically concatenations of images from MNIST and CIFAR-10.

## 1.2 NOTATIONS

We use bold lowercase letters, e.g., $\boldsymbol{v}$, and bold capital letters, e.g., $\boldsymbol{M}$, to denote vectors and matrices, respectively. The $L_p$ norm of a vector $\boldsymbol{v}$ is denoted by $\|\boldsymbol{v}\|_p$. A sample consists of feature $\boldsymbol{x}$ and label $y$. We use $\mathbb{D}$ to denote a dataset which is sampled from distribution $\mathcal{D}$. The loss of a model $f$ on a given sample is denoted by $\ell(f(\boldsymbol{x}), y)$.

## 2 INDISCRIMINATE POISONING ATTACKS USE LINEAR SEPARABLE PERTURBATIONS

In this section, we investigate the perturbations of several advanced indiscriminate poisoning attacks. First, we introduce three approaches of poisoning attacks. Then, we visualize them with two-dimensional T-SNEs. Finally, we verify that the perturbations of four advanced attacks are almost linear separable by fitting them with linear models and two-layer neural networks.

### 2.1 THREE TYPES OF INDISCRIMINATE POISONING ATTACKS

#### 2.1.1 THE ALTERNATIVE OPTIMIZATION APPROACH

We first introduce the alternative optimization approach to generate perturbations for indiscriminate poisoning attacks. It solves the following bi-level objective,

$$
\begin{aligned}
&\underset{\{\boldsymbol{\delta}\} \in \Delta}{\arg\max} \, \mathbb{E}_{(\boldsymbol{x},y) \sim \mathcal{D}}[\ell(f^*(\boldsymbol{x}), y)], \\
&\text{s.t. } f^* \in \underset{f}{\arg\min} \sum_{(\boldsymbol{x},y) \in \mathbb{D}} \ell(f(\boldsymbol{x}+\boldsymbol{\delta}), y),
\end{aligned}
\tag{1}
$$

where $\boldsymbol{\delta}$ is a sample-wise perturbation and $\Delta$ is a constraint set of all perturbations. To put in other words, the optimal solution on perturbed data (specified by the second objective) should have a maximum loss on clean data (specified by the first objective). The perturbations are restricted to not affect the normal data utility.

Directly solving Equation (1) is intractable for deep neural networks and recent works have designed multiple approximate solutions (Feng et al., 2019; Fowl et al., 2021a; Yuan & Wu, 2021). Feng et al. (2019) use multiple rounds of optimization to generate perturbations. At each round, they first approximately optimize the second objective by updating a target model on perturbed data for a few steps. Then they approximate the first objective by updating a generator for a few steps. The outputs of the generator are used as perturbations. Another example is the Neural Tangent Generalization Attacks (NTGAs) in Yuan & Wu (2021). They rewrite the entire Equation (1) into a single objective based on the recent development of Neural Tangent Kernels (Jacot et al., 2018). Then they solve the new objective with a lightweight surrogate model.

#### 2.1.2 THE ERROR-MINIMIZING NOISE

Huang et al. (2021) propose a simple way to generate data poisoning perturbations. Instead of solving Equation (1), they use the following objective,

$$
\min_f \mathbb{E}_{(\boldsymbol{x},y) \sim \mathbb{D}}[\underset{\{\boldsymbol{\delta}\} \in \Delta}{\arg\min} \, \ell(f(\boldsymbol{x}+\boldsymbol{\delta}), y)].
\tag{2}
$$

That is to say, the perturbations are intentionally optimized to reduce the training loss. A randomly initialized model is used as a surrogate of the target model. They use multiple rounds of bi-level optimization to generate perturbations. At each round, they first train the surrogate model for a few steps to minimize the loss on perturbed data. Then they optimize the perturbations to also minimize

Table 1: Training accuracy (in %) of simple models on the perturbations of different attacks.

| Algorithm | Linear Model | Two-layer NN |
|---|---|---|
| Clean Data | 49.9 | 70.1 |
| DeepConfuse (Feng et al., 2019) | 100.0 | 100.0 |
| NTGA (Yuan & Wu, 2021) | 100.0 | 100.0 |
| Error-minimizing (Huang et al., 2021) | 100.0 | 100.0 |
| Adversarial Examples (Untargeted) (Fowl et al., 2021b) | 91.5 | 99.9 |
| Adversarial Examples (Targeted) (Fowl et al., 2021b) | 100.0 | 100.0 |

the loss of the surrogate model. They repeat the above process until the loss on perturbed data is smaller than a pre-defined threshold. The main motivation is that if the training loss is zero, then the target model will have nothing to learn as there is nothing to backpropagate.

### 2.1.3 ADVERSARIAL EXAMPLES

Instead of using bi-level objectives, Fowl et al. (2021b) show that the common objectives of adversarial examples are sufficient to generate powerful data poisoning perturbations. They use both untargeted (the left objective) and targeted adversarial examples (the right objective),

$$\arg\max_{\{\boldsymbol{\delta}\}\in\Delta} \mathbb{E}_{(\boldsymbol{x},y)\sim\mathbb{D}} \left[\ell\left(f\left(\boldsymbol{x}+\boldsymbol{\delta}\right),y\right)\right], \quad \arg\min_{\{\boldsymbol{\delta}\}\in\Delta} \mathbb{E}_{(\boldsymbol{x},y)\sim\mathbb{D}} \left[\ell\left(f\left(\boldsymbol{x}+\boldsymbol{\delta}\right),y'\right)\right], \quad (3)$$

where $y' \neq y$ is an incorrect label and $f$ is a trained model. Surprisingly, Fowl et al. (2021b) demonstrate that these simple objectives can generate perturbations that achieve state-of-the-art attack performance.

### 2.2 VISUALIZING THE PERTURBATIONS

Although the three approaches in Section 2.1 have different objectives, they all manage to generate perturbations with the same effect. Intrigued by this observation, we seek to find the underlying working principle of those approaches if there is a common pattern of different types of perturbations.

To find such a common pattern, we first visualize different types of perturbations by computing their two-dimensional t-SNEs (Van der Maaten & Hinton, 2008). We generate perturbations using DeepConfuse (Feng et al., 2019), NTGA (Yuan & Wu, 2021), error-minimizing noises (Huang et al., 2021), and adversarial examples (Fowl et al., 2021b). These four attacks achieve advanced attack performance and cover all the three approaches in Section 2.1. We use the official implementations to generate perturbations. Detailed configurations are in Appendix E.

The two-dimensional t-SNEs of DeepConfuse and error-minimizing noises are shown in Figure 2. The plots of NTGA and adversarial examples are presented in Appendix B. Surprisingly, for all the attacks considered, the perturbations for the same class are well clustered, suggesting that even linear models can classify them well. For comparison, we also compute the t-SNEs of the clean data. As shown in Figure 2, in contrast with the t-SNEs of perturbations, the projections of different classes of the clean data are mixed together, which indicates that they require a complex neural network to be correctly classified.

### 2.3 PERTURBATIONS OF POISONING ATTACKS ARE ALMOST LINEAR SEPARABLE

In order to quantify the 'linear separability' of the perturbations of different indiscriminate poisoning attacks, we fit the perturbations with simple models and examine the training accuracy. The perturbations are labeled with the labels of the corresponding target examples. The simple models include linear models and two-layer neural networks. Details can be found in Appendix E.

The results are presented in Table 1. Compared to the results on clean data, simple models can easily fit the perturbations. On all attacks considered, linear models achieve more than $90\%$ training accuracy and two-layer neural networks achieve nearly $100\%$ training accuracy. These results confirm that the perturbations of advanced indiscriminate poisoning attacks are all (almost) linear separable. We note that existing attacks against deep neural networks all use ReLU activation functions so their crafting models learn piecewise linear functions in input space. In Appendix C, we replace the ReLU

layers with Tanh layers and show the perturbations are still linearly separable. This suggests that the linear separability is not stemming from the property of ReLU.

## 2.4 CONNECTING OUR FINDINGS TO SHORTCUT LEARNING

The fact that the perturbations can be easily fitted by linear models naturally connects to a recent concept named shortcut learning (Geirhos et al., 2020). Shortcut learning summarizes a general phenomenon when any learning system makes decisions based on spurious features that do not generalize on realistic test data[4]. Shortcut features have been found in different fields. For vision tasks, Beery et al. (2018) show deep models fail to recognize cows when the grass background is removed from images, suggesting the grass background is a shortcut for predicting cows. In the field of natural language processing, Niven & Kao (2019) show language models use the strong correlation between some simple words and labels to make decisions, instead of really understanding the data.

With the presence of shortcut learning, it seems reasonable to postulate that the perturbations of existing attacks succeed by creating shortcuts to the target model. We give an illustration in Figure 3. A major difference between the perturbations of poisoning attacks and existing shortcut features is that the perturbations are of an imperceptible scale and mixed together with useful features. Since there is no direct evidence to show deep models will take this kind of shortcuts, the observation that the perturbations are linear separable may only be some superficial results of the underlying root cause. In the next section, we design experiments to confirm the postulated explanation. We synthesize imperceptible and linear separable data and show deep models are very vulnerable to such synthetic shortcuts.

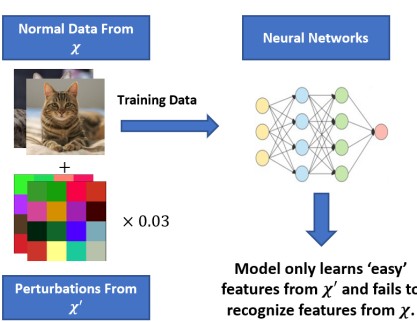

Figure 3: An illustration of the shortcut learning phenomenon in this paper.

## 3 IS LINEAR SEPARABILITY A SUFFICIENT CONDITION FOR POISONING ATTACKS TO SUCCEED?

Although we have demonstrated that the perturbations of four advanced attacks are all almost linear separable, it is a bit early to claim that 'linear separability' is the underlying working principle of indiscriminate poisoning attacks. For example, perturbations are linear separable may only be a necessary but not sufficient condition for poisoning attacks to succeed. In order to verify this postulated explanation, we use simple synthetic data to serve as perturbations and compare their effectiveness with existing poisoning attacks. It turns out that the synthetic perturbations are as powerful as the perturbations of advanced attacks.

The rest of this section is organized as follows. In Section 3.1, we first give an algorithm for generating synthetic data as perturbations. In Section 3.2, we verify the effectiveness of synthetic perturbations on different models and datasets.

### 3.1 GENERATING SIMPLE SYNTHETIC DATA

The synthetic data in this section are generated via two building blocks. In the first block, we use an algorithm in Guyon (2003) to generate samples from some normal distributions. In the second block, we transfer the samples into two-dimensional images in order to apply them to benchmark datasets.

The first building block proceeds as follows. We first generate some points that are normally distributed around the vertices of a hypercube. The points around the same vertex are assigned with the same label. Then for each class, we introduce different covariance. Any two classes of the

---

[4]Geirhos et al. (2020) use a more specific definition of shortcuts. They denote shortcuts as those features that do not generalize on out-of-distribution (OOD) data. We note that poisoning attacks would change the distribution of training data and hence make the clean test data 'OOD' with respect to the trained model.

generated points can be easily classified by a hyperplane as long as the side length of the hypercube is reasonably large. We give the pseudocode of this block in Appendix A.

---

**Algorithm 1:** Transform A 1-D Sample Into A 2-D Image

1: **Input:** input vector $\boldsymbol{x} \in \mathbb{R}^d$, target image size $s$.

   //*For simplicity, we assume the image is square and $s$ is divisible by $\sqrt{d}$.*
2: Compute patch size $p = s/\sqrt{d}$.
3: Get $\boldsymbol{x}' \in \mathbb{R}^{s^2}$ by duplicating each dimension of $\boldsymbol{x}$ for $p^2$ times.
4: Rearrange $\boldsymbol{x}'$ to get $d$ patches so that the pixel values in each patch are the same.

---

In the second building block, we pad each dimensional of the sampled points and reshape them into two-dimensional images. The padding operation introduces local correlation into the synthetic images. Local correlation is an inherent property of natural images. In Appendix F, we show the padding operation is necessary to make the synthetic perturbations remain effective when data augmentation methods are applied. The pseudocode of this block is given in Algorithm 1.

The synthetic images are scaled down before being used as perturbations. We visualize the synthetic perturbations and corresponding perturbed images in Figure 4. We also visualize the perturbations in Huang et al. (2021) for a comparison. The details of perturbations can be found in Section 3.2. As shown in Figure 4, the synthetic perturbations do not affect data utility.

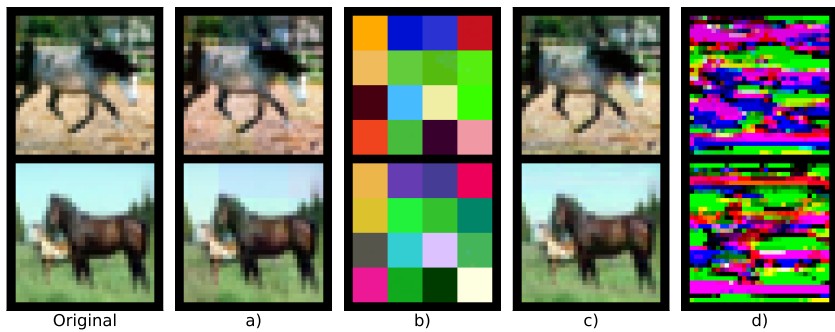

Original      a)      b)      c)      d)

Figure 4: Visualization of perturbed images and normalized perturbations. Columns a) and b) use synthetic perturbations. Columns c) and d) use the attack in Huang et al. (2021).

## 3.2 SYNTHETIC PERTURBATIONS ARE HIGHLY EFFECTIVE AS DATA POISONING ATTACK

Now we verify the effectiveness of synthetic perturbations and make comparisons with existing poisoning attacks. We perturb the entire training set following the setup in previous works (Feng et al., 2019; Huang et al., 2021; Yuan & Wu, 2021; Fowl et al., 2021b)[5]. That is, we synthesize a perturbation for every training example. We use $L_2$-norm for synthetic perturbations to keep the sample-wise variation in the same class. We normalize the synthetic noises into a $L_2$-norm ball with radius $\sqrt{d}\epsilon'$, where $d$ is the dimension of the input.

We evaluate synthetic perturbations on three benchmark datasets: SVHN (Netzer et al., 2011), CIFAR-10, and CIFAR-100 (Krizhevsky & Hinton, 2009). The target model architectures include VGG (Simonyan & Zisserman, 2014), ResNet (He et al., 2016), and DenseNet (Huang et al., 2017). We adopt standard random cropping and flipping as data augmentation. The hyperparameters for training are standard and can be found in Appendix E. We use $\epsilon' = 6/255$ for synthetic perturbations. The patch size in Algorithm 1 is set as $8$.

We first compare synthetic perturbations with existing poisoning attacks. The comparisons are made on the CIFAR-10 dataset with ResNet-18 as the target model. The perturbations in previous works

---

[5]In Appendix D, we show synthetic perturbations are still effective when different percentages of training data are poisoned (range from 20% to 90%).

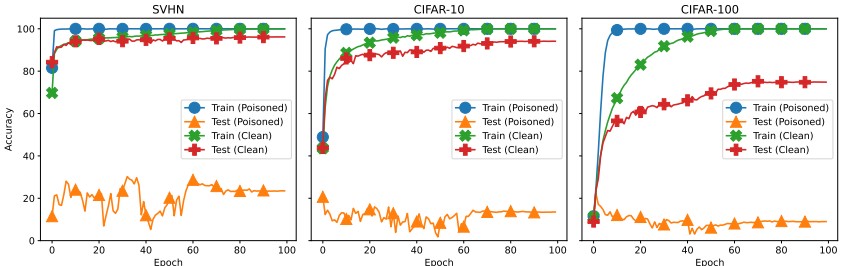

Figure 5: Training curves of ResNet-18 models on perturbed and clean data. The word 'poisoned' denotes the model is trained on perturbed data. The test performance is evaluated on clean data. The test accuracy is low throughout training when synthetic perturbations are added.

Table 2: Accuracy on clean test data of CIFAR-10. The target model is ResNet-18. The training data are poisoned with different attacks. The smaller the accuracy, the better the attack efficiency.

| Algorithm | Test Accuracy (in %) |
|---|---|
| No Perturbation | 94.69 |
| TensorClog (Shen et al., 2019) | 48.07 |
| Alignment (Fowl et al., 2021a) | 56.65 |
| DeepConfuse (Feng et al., 2019) | 28.77 |
| NTGA (Yuan & Wu, 2021) | 33.29 |
| Error-minimizing (Huang et al., 2021) | 19.93 |
| Adversarial Examples (Fowl et al., 2021b) | 6.25 |
| Synthetic Perturbations | 13.54 |

are normalized with $L_\infty$ bound with $\epsilon = 8/255$. We present the comparisons in Table 2. Then we evaluate synthetic perturbations on different models and datasets. The final test accuracy of target models is in Table 3. We also plot the training curves of target models on both clean and perturbed data in Figure 5.

As shown in Table 2, synthetic perturbations are as powerful as advanced poisoning attacks and reduce the test accuracy close to that of random guessing. The results in Table 3 and Figure 5 further confirm the effectiveness of synthetic perturbations. Notably, generating synthetic perturbations is data irrelevant and only takes several seconds using a single CPU core. We compare the computational complexities of synthetic perturbations and recent attacks in Appendix A.

In summary, our experiments demonstrate that using linear separable perturbations is indeed a sufficient condition for indiscriminate poisoning attacks to succeed. Moreover, these results also expose that deep models are very vulnerable to obscured shortcuts. This finding has two meanings to the community. First, it confirms that advanced indiscriminate poisoning attacks do succeed by providing shortcuts. Second, it further exposes the shortcut learning problem, which is a fundamental vulnerability of deep models.

Table 3: Accuracy (in %) on clean test data. The target models are trained on clean data ($\mathbb{D}_c$) and data perturbed by synthetic perturbations ($\mathbb{D}_{syn}$).

| Target Model | SVHN | | CIFAR-10 | | CIFAR-100 | |
|---|---|---|---|---|---|---|
| | $\mathbb{D}_c$ | $\mathbb{D}_{syn}$ | $\mathbb{D}_c$ | $\mathbb{D}_{syn}$ | $\mathbb{D}_c$ | $\mathbb{D}_{syn}$ |
| VGG-11 | 95.4 | 18.1 | 91.3 | 28.3 | 67.5 | 10.9 |
| ResNet-18 | 96.2 | 8.0 | 94.7 | 13.5 | 74.8 | 9.0 |
| ResNet-50 | 96.4 | 7.8 | 94.8 | 14.9 | 75.2 | 8.4 |
| DenseNet-121 | 96.7 | 9.7 | 95.0 | 10.6 | 76.5 | 7.6 |

# 4 DEFEND INDISCRIMINATE POISONING ATTACKS WITH PRE-TRAINED FEATURE EXTRACTORS

We now explore possible defenses against indiscriminate poisoning attacks with our understanding. The experiments in Section 2 and 3, suggest that deep models only use shortcuts to make predictions and ignore the useful features. A reasonable defense should be able to filter out such shortcuts and make the decision based on normal features. To achieve this goal, we explore using pre-trained models to extract useful features. Pre-training is a popular technique that trains the model on some auxiliary data before learning on target data. If the auxiliary data is similar to the target data, the pre-trained models can recognize some useful features and hence avoid relying on shortcuts.

Our implementation is as follows. We use a ResNet-152 model pre-trained on unlabeled ImageNet with SimCLR (Chen et al., 2020). To apply the pre-trained model on CIFAR-10, we remove the final layer and add a ten-class readout layer. The output dimension of the penultimate layer is $4096$. We try two different approaches to take advantage of the pre-trained model. The first approach uses the pre-trained weights as initialization and fine-tunes the whole model. The second approach only uses the outputs of the penultimate layer and trains a linear classifier on them.

We run experiments on five types of perturbed CIFAR-10 datasets that are crafted via DeepConfuse (Feng et al., 2019), NTGA (Yuan & Wu, 2021), error-minimizing noise (Huang et al., 2021), error-maximizing noise (Fowl et al., 2021b), and synthetic perturbations. The strength of perturbations is the same as that in Section 3. In Appendix G, we increase $\epsilon$ up to 32 and show the proposed defense remains effective. The hyperparameter choices of fine-tuning can be found in Appendix E.

Our baseline method is adversarial training. It is a powerful defense against existing attacks (Huang et al., 2021; Tao et al., 2021; Fowl et al., 2021b). We use Fast Gradient Sign Method (FGSM) (Goodfellow et al., 2014) with the $L_\infty$ norm bound to generate adversarial examples. We choose $\epsilon$ from a list $[2/255, 4/255, 8/255]$ and report the best test accuracy for adversarial training.

Table 4: Test accuracy (in %) when different countermeasures are applied. 'Linear eval.' means we fix the pre-trained weights and only train the classification layer.

| Algorithm | Adv. Training | Pre-train | Pre-train (Linear eval.) |
|---|---|---|---|
| No Perturbation | 93.8 | 95.9 | 95.4 |
| DeepConfuse (Feng et al., 2019) | 86.5 | 17.6 | **94.1** |
| NTGA (Yuan & Wu, 2021) | 90.8 | 39.5 | **95.2** |
| Error-minimizing (Huang et al., 2021) | 91.2 | 21.6 | **94.0** |
| Error-maximizing (Fowl et al., 2021b) | 85.4 | 37.3 | **87.6** |
| Synthetic Perturbations | 87.8 | 15.7 | **93.8** |

The results are shown in Table 4. When fine-tuning the full models, the models still pick the shortcuts and achieve poor test accuracy. In contrast, when we freeze the pre-trained weights and only train linear classifiers on the extracted features, pre-training substantially increases the test accuracy. Moreover, the pre-trained feature extractor achieves better defense than adversarial training.

Although the proposed defense achieves state-of-the-art results against advanced attacks, it could have potential risks when facing white-box attacks. The four advanced poisoning attacks evaluated in this section are all black-box, i.e., they do not have access to the target model or try to recover the target model first. The power of our defense may be compromised if the perturbations are generated to invalid the chosen pre-trained model.

# 5 CONCLUSION

This work gives an explanation of the working principle of indiscriminate poisoning attacks. We show advanced attacks coincidentally generate linear separable perturbations. We further synthesize linear separable perturbations to demonstrate that using linear separable perturbations is a sufficient condition for attacks to succeed. Our findings also suggest deep models are more prone to shortcuts than previously believed as they will find and heavily rely on shortcuts even when the shortcuts are scaled down to an imperceptible magnitude. Finally, our explanation also motivates us to use pre-trained feature extractors as a powerful defense.

## REPRODUCIBILITY STATEMENT

We upload our source code in the supplementary materials. The README files in the uploaded code provide example commands to verify our results. Our implementation is based on Pytorch (`https://pytorch.org/`), which is a popular open-source machine learning framework.

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

## A    GENERATE SYNTHETIC DATA FROM NORMAL DISTRIBUTIONS

We adopt an algorithm in Guyon (2003) to generate synthetic data from normal distributions. Algorithm 2 shows the pseudocode. The algorithm places data of different classes at different vertices of a hypercube. It also introduce different random covariance to each class.

---

**Algorithm 2:** Generating Synthetic Data From Normal Distributions

---

1: **Input:** number of classes $k$, number of examples in each class $\{n_i\}_{i=1}^k$, data dimension $d$.

2: Create a $d$-dimensional hypercube.
3: **for** $i = 1$ **to** $k$ **do**
4:     Generate $\boldsymbol{D}^{(i)} \in \mathbb{R}^{n_i \times d}$, where each row of $\boldsymbol{D}^{(i)}$ is sampled from $\mathcal{N}(0, \boldsymbol{I}_{d \times d})$.
5:     *// Introduce random covariance among columns.*
6:     Uniformly sample the elements of $\boldsymbol{A} \in \mathbb{R}^{d \times d}$ from $[-1, 1]$.
7:     Compute $\boldsymbol{D}^{(i)} = \boldsymbol{D}^{(i)} \boldsymbol{A}$.
8:     Randomly choose an unused vertex and let $\boldsymbol{c}^{(i)} \in \mathbb{R}^d$ be its coordinates.
9:     *// Move the sampled points to the chosen vertex.*
10:     Compute $\boldsymbol{D}^{(i)} = \boldsymbol{D}^{(i)} + \boldsymbol{c}^{(i)}$, i.e., $\boldsymbol{c}^{(i)}$ is added to each row of $\boldsymbol{D}^{(i)}$.
11:     Assign the rows of $\boldsymbol{D}^{(i)}$ with label $i$.
12: **end for**

---

The computational complexity of generating synthetic perturbations is $\mathcal{O}(nd/p^2)$, where $n$ is the size of dataset, $d$ is the dimension of clean data, and $p$ is the patch size in Algorithm 1. This complexity is mainly from introducing covariance into synthetic data (Line 6 in Algorithm 2). We note that the complexity of generating synthetic perturbations is significantly smaller than that of recent attacks. The complexity of running the algorithms in recent attacks is $\mathcal{O}(TLn(dw + w^2))$, where $T$ is the number of iterations generating the poisons, $L$ is the network depth, and $w$ is the network width (the cost of one forward and backward process). This complexity is strictly worse than that of generating synthetic perturbations.

## B    ADDITIONAL T-SNE PLOTS

Here we plot the t-SNEs of two other attacks in Table 1, i.e., adversarial examples (Fowl et al., 2021b) and NTGA (Yuan & Wu, 2021). We use their official implementations to generate the perturbations (see Appendix E for details). The t-SNEs are plotted in Figure 6. The perturbations for the same class are well clustered. This observation is similar to that from Figure 2.

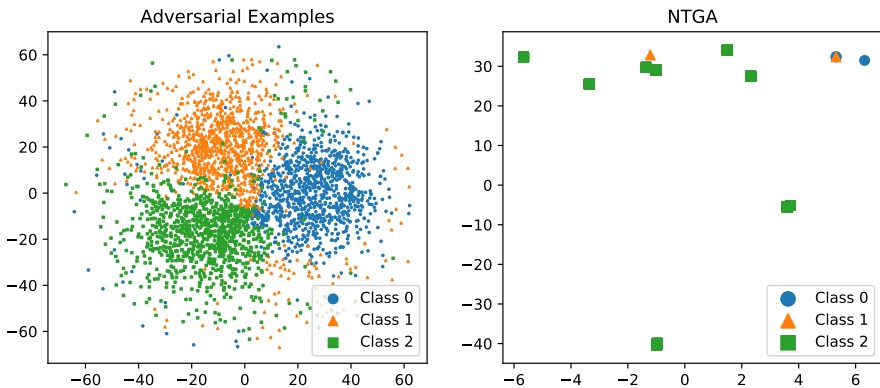

Figure 6: T-SNEs of targeted adversarial examples (Fowl et al., 2021b) and NTGA (Yuan & Wu, 2021). Perturbations from the same class are well clustered. Notably, many embeddings of NTGA are overlapped, suggesting that it uses very similar perturbations for some examples.

Table 5: Training accuracy (in %) of simple models on the perturbations of different attacks. The perturbations are generated with **Tanh-DNNs.**

| Algorithm | Linear Model | Two-layer NN |
|---|---|---|
| Error-minimizing (Huang et al., 2021) | 100.0 | 100.0 |
| Adversarial Examples (Untargeted) (Fowl et al., 2021b) | 92.7 | 100.0 |
| Adversarial Examples (Targeted) (Fowl et al., 2021b) | 100.0 | 100.0 |

Table 6: Test accuracy (in %) with different poisoning percentages $p$. Training with the poisoned subset does not improve the test accuracy much compared to training with clean data only.

| Method | $p = 90\%$ | $p = 80\%$ | $p = 50\%$ | $p = 20\%$ |
|---|---|---|---|---|
| Clean Data Only (100%-$p$) | 82.6 | 86.5 | 92.4 | 93.9 |
| Error-minimizing (Huang et al., 2021) | 85.2 | 86.8 | 92.8 | 94.1 |
| Adversarial Examples (Fowl et al., 2021b) | 85.3 | 88.2 | 92.2 | 93.7 |
| Synthetic Perturbations | 85.7 | 86.3 | 92.9 | 94.0 |

## C  LINEAR SEPARABILITY IS NOT STEMMING FROM ReLU

Existing indiscriminate poisoning attacks against deep neural networks all use ReLU activation functions in their crafting models. It is well known to the community that a ReLU-DNN learns a piecewise linear function in input space (Arora et al., 2016). To verify that whether the linear separability of perturbations is stemming from the property of ReLU, we replace the ReLU layers with Tanh layers in the crafting models of error-minimizng noises (Huang et al., 2021) and adversarial examples (Fowl et al., 2021b). We fit the new perturbations with the same simple models as those in Section 2. The results are presented in Table 5. The new perturbations are still almost linearly separable: linear models achieve more than 90% training accuracy and two-layer neural networks achieve 100% training accuracy. This suggests the linear separability is not stemming from the property of ReLU.

## D  POISONING DIFFERENT PERCENTAGES OF THE TRAINING DATA

In this section, we show synthetic perturbations are as effective as advanced attacks when only a given percentage of the training data is poisoned. We follow the method in Huang et al. (2021); Fowl et al. (2021b). For each poisoning percentage, we train two models. One model uses both the clean subset and the poisoned subset as its training data and the other one only uses the clean subset. The difference between the performances of those two models represents how much information the former model gains from the poisoned data.

We test four different poisoning percentages (from 20% to 90%) on the CIFAR-10 dataset. The experiments are run on ResNet-18 models. We compare the performance of synthetic perturbations with adversarial examples and error-minimizing noises (Huang et al., 2021; Fowl et al., 2021b). The results are presented in Table 6. The performance gain of using the poisoned subset is small for all three attacks. This suggests that synthetic perturbations are still effective in this setting.

## E  IMPLEMENTATION DETAILS OF EXPERIMENTS

**Implementation details of the experiments in Section 2.** We generate perturbations for the CIFAR-10 dataset using the official implementations of DeepConfuse[6], NTGA[7], error-minimizing noise[8], and adversarial examples[9]. The configuration is set to be the one that achieves the best attack performance on CIFAR-10. Specifically, DeepConfuse uses an 8-layer U-Net (Ronneberger et al., 2015) as the

---

[6]https://github.com/kingfengji/DeepConfuse
[7]https://github.com/lionelmessi6410/ntga
[8]https://github.com/HanxunH/Unlearnable-Examples
[9]https://github.com/lhfowl/adversarial_poisons

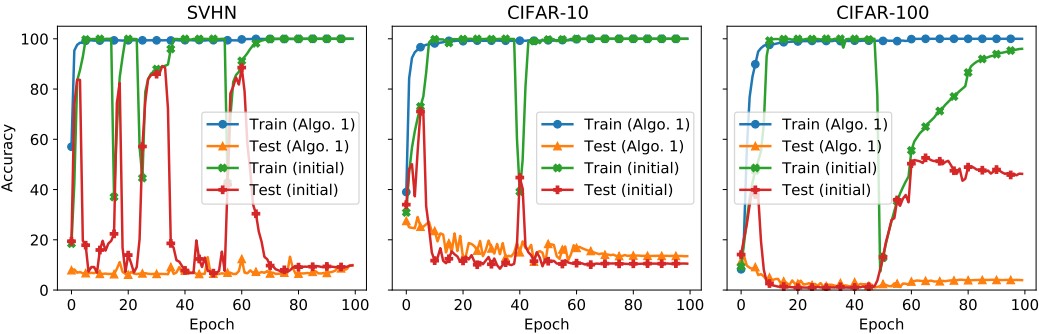

Figure 7: Training curves of ResNet18 models trained on SVHN, CIFAR-10, and CIFAR-100 datasets. The word "initial" denotes the synthetic perturbations are generated directly using Algorithm 2.

crafting model. NTGA uses a 3-layer convolutional network. Error-minimizing noises and adversarial examples use standard ResNet-18 models.

The experimental setup for training the simple models are as follows. We train the simple models with standard cross-entropy loss. Before training, all perturbations are flattened into 1-dimensional vectors and normalized to unit norm. The two-layer neural networks have a width of 30. All models are trained with the L-BFGS optimizer (Liu & Nocedal, 1989) for 50 steps.

**Implementation details of the experiments in Section 3.** We use the Stochastic Gradient Descent (SGD) optimizer with a momentum coefficient 0.9 for all experiments. For all datasets, we use a batchsize of 128. The learning rates of all models are set to follow the choices in the original papers (Simonyan & Zisserman, 2014; He et al., 2016; Huang et al., 2017). The learning rate for ResNet and DenseNet models is 0.1. The learning rate for VGG models is 0.01. All models are trained for 100 epochs. The learning rate is divided by 10 at epoch 50 and 75.

**Implementation details of the experiments in Section 4.** To fine-tune the pre-trained model on the CIFAR-10 dataset, we first use bicubic upsampling to resize the image into $224 \times 224$. For the first approach, we use the same hyperparameters as Section 3 to fine-tune the full model. For the second approach, we first extract the features of all samples and train linear classifiers on the extracted features. The linear classifiers are trained with the L-BFGS optimizer for 50steps.

## F    THE EFFECT OF THE PADDING OPERATION IN ALGORITHM 1

Here we explain why we duplicate each dimension of the initial data points into two-dimensional patches in Algorithm 1. Intuitively, it is more convenient to directly generate synthetic perturbations that have the same dimension as the original images. We will show this straightforward approach has unstable performance when common data augmentation methods are applied.

We implement the above straightforward by directly using the output of Algorithm 2, i.e., the dimension of synthetic data is the same as the dimension of flattened images and we simply reshape the synthetic data into two-dimension. Other configurations are the same as those in Section 3. The models are trained with standard augmentation methods including random crop and flipping. We compare this straightforward approach with the one that further processes the outputs with Algorithm 1. The training curves of the target models are plotted in Figure 7. When using the straightforward approach, the test accuracy sometimes increases to a high point which violates the requirement of indiscriminate poisoning attacks.

## G    INCREASING THE STRENGTH OF PERTURBATIONS

In Section 4, we only use a single level of perturbation. Here we increase the perturbation strength and examine whether the proposed defense remains effective. Specifically, we try $\epsilon = 16, 32$ for existing attacks and $\epsilon' = 12, 24$ for synthetic perturbations. Other settings are the same as those in Section 4. The results are presented in Table 7. Even when strong perturbations are applied ($\epsilon = 32$

Table 7: Test accuracy (in %) when the proposed defense is applied. We fix the pre-trained weights and only train the linear readout layer.

| Algorithm | $\epsilon = 16$ ($\epsilon' = 12$) | $\epsilon = 32$ ($\epsilon' = 24$) |
|---|---|---|
| DeepConfuse (Feng et al., 2019) | 92.4 | 82.4 |
| NTGA (Yuan & Wu, 2021) | 92.9 | 84.4 |
| Error-minimizing (Huang et al., 2021) | 92.2 | 87.9 |
| Error-maximizing (Fowl et al., 2021b) | 82.6 | 76.6 |
| Synthetic Perturbations | 92.0 | 87.4 |

and $\epsilon' = 24$), the proposed defense still achieves decent test accuracy. We note that increasing the perturbations strength will make the perturbations visible. This will ultimately violate the designing goal of indiscriminate poisoning attacks in this paper that is to reduce the test accuracy without hurting the normal data utility.

