# OpenReview forum: "Indiscriminate Poisoning Attacks Are Shortcuts"
_ICLR.cc/2022/Conference — ICLR 2022 Submitted_

### Official Review · Reviewer_ngSk · 2021-10-29

**Correctness:** 3
**Technical Novelty And Significance:** 4
**Empirical Novelty And Significance:** 4
**Recommendation:** 8
**Confidence:** 5

**Main Review:**

Strengths:

1. This paper shows that linear separability is indeed an underlying principle across many delusive attacks. I thought this finding is very interesting and would help the community to better understand the mechanisms behind the attacks.
2. The proposed synthetic noise is very effective, which makes the finding about the linear separability more solid.
3. The connection between linear separability and shortcut learning is thought-provoking. This shortcoming about shortcuts may also be related to the simplicity bias of neural networks, as observed in [3].

Weaknesses:

1. Some citations in the fourth sentence in the introduction section are not appropriate. Specifically, Fawkes (Shan et al., 2020) and LowKey (Cherepanova et al., 2021) are actually targeted attacks (i.e., causing test-time error on some given samples), rather than indiscriminate attacks (i.e., causing test-time error on all unseen samples) in the context.
2. The criticism from Radiya-Dixit & Tramer (2021) may fail to challenge the effectiveness of the indiscriminate attacks considered in this paper, since their criticism was directed only at those targeted attacks such as Fawkes and LowKey.
3. The first sentence in the abstract describing the indiscriminate attacks is somewhat imprecise. Specifically, not all indiscriminate data poisoning attacks work by adding imperceptible perturbations. Actually, another type of indiscriminate attacks can increase test error by injecting a small amount of arbitrarily crafted examples into the training set (e.g., [1]). Both two types of poisoning attacks belong to indiscriminate attacks. Therefore, it would be better to call the former "clean-label indiscriminate attacks" or, simply, "delusive attacks" as in [2-3].
4. The last sentence on page 1 is imprecise. Specifically, it was [4] that pointed out for the first time that "adversarial examples can serve as an indiscriminate poisoning attack". Later, [3] and [5] both experimented with this idea. The difference is that [3] only verified targeted adversarial examples, while [5] further tested untargeted adversarial examples.
5. In Table 1, it is suggested to report the results of the linear separability for both targeted and untargeted adversarial examples [3-5]. *The content in Section 2.1.3 may also need to be modified to precisely reflect this subtle difference.*
6. In Table 4, it is suggested to further report the performance of adversarial training with small $\epsilon$ as baselines, since it has been found that adversarial training with smaller $\epsilon$ may have better defense performance [3].

[1] Biggio et al., Poisoning Attacks against Support Vector Machines, ICML 2012
[2] Newsome et al., Paragraph: Thwarting Signature Learning by Training Maliciously, RAID 2006
[3] Tao et al., Better Safe Than Sorry: Preventing Delusive Adversaries with Adversarial Training, NeurIPS 2021
[4] Nakkiran, "A Discussion of 'Adversarial Examples Are Not Bugs, They Are Features': Adversarial Examples are Just Bugs, Too", Distill, 2019.
[5] Fowl et al., Adversarial Examples Make Strong Poisons, NeurIPS 2021
[6] Huang et al., Unlearnable Examples: Making Personal Data Unexploitable, ICLR 2021

**Summary Of The Paper:**

This paper unveils an intriguing "linear separable" property of various existing indiscriminate poisoning attacks, and then designs a new attack using linearly separable features. Experiments show that the proposed attack is fast and efficient. Additionally, pre-trained feature extractors are suggested as a powerful defense strategy.

**Summary Of The Review:**

Overall, I feel that the finding of linear separability would be of great interest to the community, and I consider the proposed synthetic noise as a supporting evidence for the main claim about the underlying principle and as a direct application of the linear separability. Therefore, I am leaning to accept. I would like to increase my score if the concerns are well addressed.

---

> ### Author Response · Authors · 2021-11-19
> **Response to Reviewer ngSk**
>
> Thank you for your review and constructive comments. Please find our response below. We have updated our submission according to the response. We use blue text to highlight the changes.
>
> **Q1**: Citations of Fawkes and LowKey are not appropriate.
>
> A1: In the updated version, we move their citations to the related work and introduce them as attacks that cause test-time error on some given samples, i.e., the face images of one specific person.
>
> **Q2**: The criticism from Radiya-Dixit & Tramer (2021) was directed only at targeted attacks.
>
> A2: The general idea that the adversary can simply store the poisoned data and wait for better defenses may also apply to indiscriminate attacks. In the updated version, we state that Radiya-Dixit & Tramer (2021) only verify the above idea is achievable for targeted attacks.
>
> **Q3**:  The first sentence in the abstract describing the indiscriminate attacks is somewhat imprecise.
>
> A3: We add a footnote to the first sentence to clarify that we only focus on indiscriminate attacks that add imperceptible perturbations to existing samples although there are some attacks injecting malicious training samples. We also clarify that this type of attack is also known as “delusive attack” or “availability attack”.
>
> **Q4**: It was Nakkiran (2019) that pointed out for the first time that “adversarial examples can serve as an indiscriminate poisoning attack”.
>
> A4: Thanks for the clarification. In the revised version, we add references to [3,4,5] in chronological order.
>
> **Q5**: It is suggested to report the results of the linear separability for both targeted and untargeted adversarial examples.
>
> A5: We add the results of targeted adversarial examples in Table 1. We use the official implementation of [5] to generate the perturbations. Both the linear model and two-layer NN achieve 100% training accuracy on targeted perturbations. We also revise Section 2.1.3 to include the introduction of targeted adversarial examples.
>
> **Q6**: It is suggested to further report the performance of adversarial training with small $\epsilon$ as baselines.
>
> A6: In the revised version, we choose $\epsilon$ from [2/255, 4/255, 8/255] for adversarial training and report the best accuracy in Table 4. Although smaller $\epsilon$ improves the test accuracy of adversarial training by roughly 5% in average, the average test accuracy of the proposed defense is still better. The average test accuracy of our defense is still 4.6% higher than that of adversarial training.

---

> > ### Comment · Reviewer_ngSk · 2021-11-27
> > **Thanks**
> >
> > Thanks for the revision and detailed rebuttal. I think the clarity is largely improved now. A minor point is that the algorithm of targeted adversarial examples should be attributed to Nakkiran [4], though its SOTA performance is achieved with additional experimental tricks in later works (e.g., 8.69% in Table 2 of [5], and 9.40% in Table 2 of [3]).
> >
> > Nevertheless, I do believe the discovery of linear separability is valuable to the adversarial ML community and thus will have a good impact. Thus, I am still leaning to accept this paper.

---

> > > ### Author Response · Authors · 2021-11-28
> > > **Continued Response**
> > >
> > > Thanks for the new comments. In the next revision, we will clarify Nakkiran [4] first propose the algorithms of targeted adversarial examples but the tricks in later works are necessary to make the algorithm a strong attack (which is the baseline in our experiments).

---

### Official Review · Reviewer_nWuM · 2021-10-31

**Correctness:** 3
**Technical Novelty And Significance:** 3
**Empirical Novelty And Significance:** 3
**Recommendation:** 8
**Confidence:** 5

**Main Review:**

Overall, I lean towards recommending accepting this paper.

---
Strengths:

- The understanding of this paper is interesting. The understanding applies to a wide range of existing works in this area. The indiscriminate data poisoning attacks are a form of linear separable features. The experimental results (t-SNE and shallow models) supports this claim.
- Based on this understanding, the proposed synthesized linear separable data is technically sound and shows a effective performance.
- This paper's presentation is easy to follow and most related works have been discussed.

---

Question/weakness:
- Are other attacks evaluated in Table 1/2 also showing linear separable on t-SNE?
- The experiments baseline is not consistent. For tables 2 and 4, why not include Alignment/TensorClog in table 4?
- Legends and captions in Figure 4 are not informative enough. What does test (poisoned) mean? Is it training on poisoned training data evaluated against clean test data? Or is it the poisoned test data?
- Need clarifications in the shortcut features. At the beginning of section 2.4, you describe shortcut learning as spurious features that are useful for training but do not generalize to test data. I believe in Geirhos et al., 2020, it is described as do not generalize to the OOD test set, but these features do generalize to the IID test set. I understand the connection of indiscriminate poisoning attacks, but some evidence/figures would be very helpful to establish this connection.



**Summary Of The Paper:**

This paper demonstrated existing indiscriminate data poisoning attacks are a form of linear separable features. They exploit the shortcut learning of deep learning. Based on the understanding, this paper also shows a simple synthetic data can be used to achieve the same effect as existing indiscriminate data poisoning attacks.

**Summary Of The Review:**

This paper made an interesting observation on the linear separability of a wide range of indiscriminate data poisoning attacks. This reveals the reason why such attacks can succuss.

---

> ### Author Response · Authors · 2021-11-19
> **Response to Reviewer nWuM**
>
> Thank you for your review. Please find our response below. We have also updated our submission according to the response. We use blue text to highlight the changes.
>
> **Q1**: Are other attacks evaluated Table 1/2 also showing linear separable on t-SNE?
>
> A1: We add the t-SNE plots of adversarial examples and NTGA in Appendix B. For NTGA, we download the perturbations from [1]. For adversarial examples, we use the command in [2] that gives the most powerful perturbations. The perturbations for the same class are also well clustered for these two attacks. Notably, many embeddings of NTGA are overlapped, suggesting that it uses very similar perturbations for some examples.
>
> **Q2**: Why not include Alignment/TensorClog in Table 4?
>
> A2: We do not include Alignment and TensorClog in Table 1 and Table 4 because we do not have their perturbations in our hand (we do not find off-the-shelf implementations for them). The numbers in Table 2 are directly from their papers. The four attacks we implement are representative: they achieve advanced attack performance and cover all the three types of attacks in Section 2.1.
>
> **Q3**: What does test (poisoned) in Figure 4 mean?
>
> A3: It means training on poisoned training data and evaluating on clean test data. We have clarified this in the caption in the revised draft.
>
> **Q4**:  Need clarifications for the shortcut features. In Geirhos et al., 2020, the shortcut features do generalize to the IID data.
>
> A4: In the updated version, we clarify the definition of shortcuts in Geirhos et al., 2020. We also explain more about how the shortcuts in Geirhos et al., 2020 are connected to existing attacks. The perturbations will change the distribution of training data. If the model only learns the artificial shortcuts, the normal ‘IID’ test data will be ‘OOD’ for the trained model. In the revised version, we add Figure 3 to illustrate the shortcut features in this paper.
>
> [1]: https://github.com/lionelmessi6410/ntga. Neural Tangent Generalization Attacks (NTGA).
>
> [2]: https://github.com/lhfowl/adversarial_poisons. Adversarial poison generation and evaluation.
>
> We expect the reviewer would consider increasing the score if the concerned points have been well addressed given that the reviewer is confident about the value of our contribution.

---

> > ### Author Response · Authors · 2021-11-28
> > **Continued Response**
> >
> >
> >
> >
> > Hi Reviewer nWuM, We have posted our responses and revised the submission accordingly. Does our response address your concerns, or are there other aspects that you think would help improve the paper? Thanks!

---

> > > ### Comment · Reviewer_nWuM · 2021-11-28
> > > **Thanks for addressing concerns.**
> > >
> > > The TensorClog implementation is publicly available on GitHub. I believe the Alignment uses code from Witches’ Brew (Geiping et al., 2020) which is also available on GitHub. If this paper is accepted, for completeness, includes them for the t-SNE figure and Table 1.
> > >
> > > After reading the response and revision, my previous concerns/questions are addressed. I do believe this understanding is beneficial to the adversarial machine learning community, thus has increased my score.
> > >
> > > ---
> > > https://github.com/JonasGeiping/poisoning-gradient-matching
> > >
> > > https://github.com/JC-S/TensorClog_Public

---

> > > > ### Author Response · Authors · 2021-11-30
> > > > **Thanks for the new comments and references!**
> > > >
> > > > We are using the code from Witches’ Brew (Geiping et al., 2020) to generate perturbations for Alignment. We will also contact the authors of TensorClog to get instructions to run their code.

---

### Official Review · Reviewer_baCa · 2021-11-02

**Correctness:** 4
**Technical Novelty And Significance:** 1
**Empirical Novelty And Significance:** 2
**Recommendation:** 3
**Confidence:** 5

**Main Review:**

The finding that existing attacks produce linearly-separable datasets is interesting and somewhat counters the standard explanation offered as to why these attacks are effective.

At the same time, the rest of the paper's contributions are rather minor:
- The finding that shortcuts are sufficient for poisoning is relatively well-known. There has already been work that shows that if the training set contains a class-indicative, easy-to-learn pattern the classifier will disproportionately rely on that (e.g., [DecoyMNIST](https://arxiv.org/abs/1703.03717) or [https://arxiv.org/abs/1811.00401](https://arxiv.org/abs/1811.00401) in the context of interpretability). In fact, Tao et al. 2021 (cited in the manuscript) propose a very similar yet simpler attack (P5 in their paper) with virtually the same justification.
- As the authors acknowledge, the proposed defense is only demonstrated to be effective when pre-training is performed on clean data and the adversary has no knowledge of the defense in place. However, these are relatively strong assumptions. An adversary could devise methods to circumvent the defense were this not the case.

**Summary Of The Paper:**

The authors study data poisoning attacks that aim to degrade the overall (test) accuracy of a model by adding small perturbations to all training samples. They find that existing attacks introduce features into the data that make the training dataset easy to fit without learning the underlying concept (and thus not in a way that would generalize to natural test examples). They then show that this principle is sufficient to create poisoning attacks by proposing a simple method relying on random, class-specific pattern. Finally, the authors propose defending against these such poisoning attacks by relying on a pre-trained model and only fine-tuning the last (linear) layer.

**Summary Of The Review:**

While the work does contain some interesting findings, the overall contribution is rather minor when taking into account prior work. Thus, I do not find it suitable for the general NeurIPS audience.

---

> ### Author Response · Authors · 2021-11-19
> **Response to Reviewer baCa**
>
> Thank you for your review. Please find our response below.
>
> We thank the reviewer for acknowledging that the finding that existing attacks produce linearly-separable datasets is interesting. This is indeed our main contribution as indicated in the paper title.
>  This finding would be of great interest to the community as it exposes the underlying mechanism of existing complicated attacks. This finding may also reshape the direction of future works of model attack and defense in other ways.
>
> **Q1**: The finding that shortcuts are sufficient for poisoning is relatively well-known.
>
> A1: It is well known that deep models rely on easy-to-learn patterns. However, previous works have not demonstrated that shortcuts are sufficient to perform strong poisoning attacks. Both the artificial datasets in [Jacobsen et al., 2018]( https://arxiv.org/abs/1811.00401) and [Ross et al., 2017](https://arxiv.org/pdf/1703.03717.pdf) contain **visible** perturbations while the attacks studied in this work use imperceptible perturbations. In the revised version, we add a discussion with [Jacobsen et al., 2018]( https://arxiv.org/abs/1811.00401) and [Ross et al., 2017](https://arxiv.org/pdf/1703.03717.pdf). In fact, recent works [1,2,3] in top-tier conferences are still working on different algorithms and our work explains the underlying reason for these different algorithms.
>
> The P5 attack in Tao et al., 2021 cannot justify shortcuts are strong poisoning attacks **because it is not effective in the early training stage.** Both Huang et al., 2021 [1] and Tao et al., 2021 propose to use class-wise random noise as a simple attack, i.e., the P5 attack in Tao et al., 2021. Huang et al., 2021 show this simple attack is ineffective in the early training stage (see [Figure 1]( https://openreview.net/pdf?id=iAmZUo0DxC0) in Huang et al., 2021). In contrast, recent attacks and the proposed synthetic perturbations are effective throughout training. This is because we process synthetic data with Algorithm 1 (related discussion is in our Appendix F).
>
> **Q2**: The proposed defense has strong assumptions: 1) it is only effective when pre-training is performed on clean data; 2) the adversary has no knowledge of the defense in place.
>
> A2: For the first concern, we note that the adversary cannot poison the pre-training data for those models that are **already** trained. Many pre-trained models for common vision and language tasks are already publicly available. For the second concern, it is most likely that the adversary does not have white-box access to the target model in practice, e.g., recent attacks all use a black-box threat model. Therefore, it is hard to deliberately craft perturbations against a specific pre-trained model. Moreover, existing defenses, e.g., adversarial training, also assume the adversary does not know there is a defense during training.
>
> Again, this is not a defense paper. We do acknowledge that the defense could be vulnerable when facing white-box attacks refer to the paper paragraph. The defense strategy of using pretrained model naturally arises based on our understanding of the intrinsic property of data poisoning attacks. It turns out to be very effective in current setting compared to some other approaches like adversarial training, which verifies our main point from a side perspective.
>
>
> [1]: Unlearnable Examples: Making Personal Data Unexploitable. ICLR 2021, https://arxiv.org/abs/2101.04898.
>
> [2]: Neural Tangent Generalization Attacks. ICML 2021, https://proceedings.mlr.press/v139/yuan21b.html.
>
> [3]: Adversarial Examples Make Strong Poisons. NeurIPS 2021, https://arxiv.org/abs/2106.10807.

---

> > ### Comment · Reviewer_baCa · 2021-11-25
> > **Rebuttal response**
> >
> > I appreciate the authors response. Unfortunately, my main points of criticism still stand:
> > - Previous works _do_ show that random class-wise patterns are sufficient for poisoning (Tao et al., Huang et al., ++). The fact that these attacks might not work throughout training (which might in fact be an artifact of the specific attacks and not a general property), is not particularly relevant as long as they are effective by the end of training. Maybe the proposed attack is more successful and reliable, but the fact that simple class-wise perturbations work is definitely not novel given the literature.
> > - The current pre-training results are not sufficient to be considered as a major contribution. If the authors do want to claim that this method is an effective defense, then they need to at least provide some experiments on defense-aware attacks. The claim "...it is hard to deliberately craft perturbations against a specific pre-trained model." is not sufficient to justify the lack of at least some white-box evaluation.
> >
> > Thus, I still believe that the contributions are rather minor and I am keeping my score.

---

> > > ### Author Response · Authors · 2021-11-26
> > > **Continued Response**
> > >
> > > We thank the reviewer's response and appreciate the efforts of reviewing. Although we could not reach an agreement, we want to express that the reviewer may overlook our main contribution of finding perturbations are linearly separable if focusing on the two minor points.

---

> > > > ### Comment · Reviewer_baCa · 2021-11-26
> > > > **Not minor points**
> > > >
> > > > The fact that shortcut-based poisoning attacks already exist is far from a "minor" point and actually quite relevant when judging the contributions of the paper.
> > > >
> > > > If the only contribution of the paper is the linear separability of existing attacks, then I would not consider it of relevance to a wide ICLR audience.

---

> > > > > ### Author Response · Authors · 2021-11-26
> > > > > **Thanks for your timely response.**
> > > > >
> > > > >
> > > > > We do consider our main contribution is to point out that the linear separability is the workhorse of existing attacks, which we still believe would be of great interest to the community. Thanks again for your efforts and comments.

---

### Official Review · Reviewer_sUUe · 2021-11-04

**Correctness:** 3
**Technical Novelty And Significance:** 2
**Empirical Novelty And Significance:** 2
**Recommendation:** 5
**Confidence:** 4

**Main Review:**

Better understanding the root causes behind the success of indiscriminate poisoning attacks would be a valuable contribution. Unfortunately, this work in its current form presents different issues.

Ambiguous usage of the nomenclature. The authors refer to the poisoning attacks that maximize the error on the test dataset as "indiscriminate poisoning attacks." However, this may be confusing, and it may be worth clarifying the terms. Poisoning attacks that aim to maximize the test error are usually categorized as availability attacks. The term "indiscriminate" in Barreno et al. (2010) has a different meaning, i.e., that the attacker is not targeting a specific user/test sample (but rather any of them – think, e.g., to phishing vs spearphishing). In more recent papers, however, multiclass classification has been considered, and targeted/indiscriminate has been used to refer to the class which the attacker aims the test sample to be assigned to. I think here it would be good to try to disambiguate the terms – see, e.g., Munoz-Gonzalez et al., 2017.

Some claims are not supported by experimental evidence. The authors have made different claims, but unfortunately, the following are not supported experimental evidence.

1) The authors claim the poisoning samples generated adding imperceptible perturbations to the training data are well separable (accordingly to the class of the original sample) in input space with hyperplanes. However, this might be due to the type of classifier considered. For example, does this hold when the attacker targets a Support Vector Machine with an RBF kernel? The authors should consider different types of classifiers and not only deep neural networks with ReLU activation functions (ReLU-DNNs indeed learn a piecewise linear function in input space).

2) The authors claim that linear separability is necessary for indiscriminate poisoning attacks to succeed. However, this claim is not supported by evidence. The authors have only shown that the existing poisoning attack algorithms produce linearly separable perturbations when run against a deep neural network. Nevertheless, this does not exclude that poisoning samples that are not linearly separable remain effective. It may be thus worth investigating the possibility of crafting “adaptive” poisoning attacks that remain undetected when considering separating hyperplanes. Another issue may be related to the high dimensionality of the input space. Is the fact that such attacks are linearly separable due to the high dimensionality of the data, other than the given model?

3) The authors claim the proposed attack is more efficient than the state-of-the-art ones. They should add a comparison of their computational complexity.

4) The authors claim that using a pre-trained model is sufficient to defend against clean-label poisoning attacks. However, their experiments have considered a single level of perturbation that the attacker can inject. To support their claim, they should instead assess the attack effectiveness for an increasing level of perturbation the attacker can add. The same observation has also been done in this work (https://arxiv.org/abs/2106.07214) for backdoor attacks, so it may be worth clarifying that this finding is not completely novel.

Overall, the authors should present a more extensive experimental analysis to support their claims.

The experimental setting is not sufficiently described. For some experiments, the experimental details are not provided. For example, against which deep neural network have the authors crafted the poisoning samples to generate Table 1?

Confusing manuscript organization. The description of the experimental setup is fragmented into different sections.



**Summary Of The Paper:**

The authors empirically investigate the underlying reasons behind the effectiveness of indiscriminate poisoning attacks. According to the authors' claims, their analysis shows that the poisoning samples generated adding imperceptible perturbations to the training data are well separable (accordingly to the class of the original sample) in input space with hyperplanes. This discovery enables the authors to propose a more efficient attack, and to understand that using a pre-trained neural network as a feature extractor can enable defending such poisoning attacks.

**Summary Of The Review:**

The paper tries to clarify an issue, but more insights need to be developed. For instance, whether such linear separability is induced by the kind of model, input dimensionality, and if it generalizes beyond poisoning attacks – similar evidence are probably existing for backdoor poisoning and adversarial examples too.

I’m also quite concerned about the threat model, as poisoning large fractions of points (more than 20%) is not quite realistic in many practical scenarios (I don’t think it’s realistic to poison up to 50% of the dataset). This should be clarified in the paper, and more details on the practicality of the threat model should be discussed or acknowledged as specific limitations of this work (and other papers that use similar setups).

---

> ### Author Response · Authors · 2021-11-19
> **Response to Reviewer sUUe (Part 2/2)**
>
>
> **Q7**: The description of the experimental setup is fragmented into different sections.
>
> A7: We move the experimental setups of generating baseline perturbations and training simple models into Appendix E. Now the main text only contains the experimental setup about synthetic perturbations (in Section 3.2).
>
> **Q8**: Does the linear separability generalize beyond poisoning attacks?
>
> A8: For adversarial examples, we add the results of targeted adversarial examples to Table 1. Now Table 1 contains the results of fitting both untargeted and targeted adversarial examples with simple models. On untargeted adversarial examples, both the linear model and two-layer NN achieve >90% training accuracy. On targeted adversarial examples, both the linear model and two-layer NN achieve 100% training accuracy. For backdoor attacks, it is a bit tricky to test the linear separability as the triggers are often designed for one single class. Nonetheless, backdoor attacks may also be some ‘easy’ patterns that serve as shortcuts.
>
> **Q9**: The threat model is not quite realistic.
>
> A9: In Section 3.2, we clarify that our threat model perturbs 100% of the training data, which is also the main threat model in previous works. Our main aim is to understand why existing attacks work instead of designing better attacks. Therefore, we follow such a threat model in the main text.
>
> In the revised version, we add Appendix D that considers threat models with smaller poisoning percentages (range from 20% to 90%). It shows that the poisoning attacks are still effective, i.e., the model cannot learn much from the poisoned data. Moreover, our synthetic perturbations share a similar effect with existing attacks under those settings.
>
>
> [1]: Neural Tangent Generalization Attacks. ICML 2021, https://proceedings.mlr.press/v139/yuan21b.html.
>
> [2]: Learning to Confuse: Generating Training Time Adversarial Data with Auto-Encoder. NeurIPS 2019, Learning to Confuse: Generating Training Time Adversarial Data with Auto-Encoder (neurips.cc).
>
> [3]: Unlearnable Examples: Making Personal Data Unexploitable. ICLR 2021, https://arxiv.org/abs/2101.04898.
>
> [4]: Adversarial Examples Make Strong Poisons. NeurIPS 2021, https://arxiv.org/abs/2106.10807.
>
> [5]: Backdoor Learning Curves: Explaining Backdoor Poisoning Beyond Influence Functions. 2106.07214.pdf (arxiv.org).
>
> [6]: U-net: Convolutional networks for biomedical image segmentation. https://arxiv.org/abs/1505.04597.

---

> > ### Comment · Reviewer_sUUe · 2021-11-25
> > **Response to rebuttal**
> >
> > I would like to thank the authors for their clarifications and for the effort they put into improving the manuscript. I think that now their contributions are clearer, even though the work is restricted to DNN models and existing attacks (i.e., no adaptive attacks have been considered to show that the linearity assumption may be questioned/violated). Moreover, even if previously published papers consider manipulating 100% of the training set, I don't think this threat model is practical and would not encourage following this line of research. For this reason, I'm keeping my score.

---

> > > ### Author Response · Authors · 2021-11-28
> > > **Continued Response**
> > >
> > >
> > > Thanks for your new comments. In Appendix D of the revised version, we add experiments that use different poisoning percentages. Moreover, our main scope is not to follow previous works to propose a better attack but is to explain why existing attacks work (as indicated in the paper title). We believe our explanation will help the community to have a better understanding of this line of research.

---

> ### Author Response · Authors · 2021-11-19
> **Response to Reviewer sUUe (Part 1/2)**
>
> We want to first thank the reviewer for the constructive comments. Please find our response below. We have also updated our submission. We use blue text to highlight the changes.
>
> **Q1**: Ambiguous usage of the nomenclature.
>
> A1: Thanks for raising this question. Indeed, there are several names for this kind of attack. We choose ‘indiscriminate’ to describe the populational behavior of such poisoning attacks, which is aligned with the main message that the poisoning perturbations are linearly separable. From system functionality’s perspective, these attacks are also ‘availability’ attacks. In the first page of the revised version, we clarify the word ‘indiscriminate’ has a different meaning as the one in Munoz-Gonzalez et al., 2017, and this kind of attack is also known as availability attack.
>
> **Q2**: The authors should consider different types of classifiers. The perturbations are linearly separable may because the attackers use ReLU-DNNs to craft them.
>
> A2:  Thanks for raising this question. We confine our scope to the data poisoning attacks recently designed for DNNs and clarify this in the revised version.
>
> To verify whether ReLU really makes difference, we conduct the following experiments. We relace all the ReLU layers with Tanh layers in the crafting models of error-minimizing noises [3] and adversarial examples [4]. The new perturbations are still almost linearly separable: linear classifiers achieve more than 90% training accuracy and two-layer neural networks achieve 100% training accuracy. This suggests the linear separability of poisoning perturbation is not much related to the property of ReLU, but rather an intrinsic property and functional underlying reason lied in these poisoning attacks. These results have been added to Appendix C in the revised version.
>
>
> **Q3**: There may exist effective poisoning samples that are not linearly separable. && Is the fact that such attacks are linearly separable due to the high dimensionality of the data, other than the given model?
>
> A3: To state more precisely, we claim that the linear separability exists in **existing** recent poisoning attacks. We cannot exclude the future possibility that perturbations that are not linearly separable remain effective. We change the wording in the paper to avoid such confusion.
>
> The perturbations are linearly separable is not because of the dimensionality of the data. In the setting of CIFAR10, the sample sizes are 50000 while the input dimension is 3072, which is usually not viewed as a high-dimensional regime where #samples < #parameters. Moreover, in Table 1, linear models have bad accuracy on clean data, which have the same dimension as the perturbations.
>
> **Q4**: The authors should add a comparison of computational complexity.
>
> A4: The computational complexity of generating synthetic perturbations is $O(nd/p^{2})$ where $n$ is the size of dataset, $d$ is the feature dimension, and $p$ is the patch size in Algorithm 1. It is not related to the neural network size and the number of iterations. In comparison, the existing attacks require $O(TLn(dw+w^{2}))$, where T is the number of iterations generating the poisons, L is the network depth and w is the network width.
>
> We add such comparison in Appendix A, as we are not advocating new attack approaches rather than demonstrating the point that linearly separable poisons are effective for successful attacks.
>
> **Q5**: The authors only use perturbations with a single level of strength when defending against them. && A previous work has run backdoor attacks against pre-trained models.
>
> A5: In Appendix G, we use perturbations with two large levels of strength. We use $\epsilon=16,32$ for existing attacks and $\epsilon’=12,24$ for synthetic perturbations. Even when $\epsilon=32$ and $\epsilon'=24$, the proposed defense still achieves decent test accuracy (the average accuracy on five attacks is larger than 80%). We note that increasing the strength of the perturbation will make the perturbations visible. This will ultimately violate the designing goal of indiscriminate poisoning attacks that is to reduce the test accuracy without hurting the normal data utility.
>
> We add a discussion with [5] in the revised version. They are the first to explore running poisoning attacks against pre-trained feature extractors. Nonetheless, they focus on backdoor attacks and do not advocate using pre-trained models as a defense.
>
> **Q6**: What is the crafting model of the poisoning samples that are used to generate Table 1?
>
> A6: The crafting models are the same as those in the official implementations of [1,2,3,4]. DeepConfuse uses an 8-layer U-Net [6]. NTGA uses a 3-layer convolutional network. Error-minimizing and error-maximizing noises use standard ResNet-18 models. We have clarified this in Appendix E of the revised version.

---

### Decision · Program_Chairs · 2022-01-20

**Decision:**

Reject

**Comment:**

The paper studies poisoning attacks of small perturbations to training samples. Existing attacks introduce features that allow for easily fitting the data, but do not lead to good generalization. They use this principle to generate attacks based on random class-specific patterns. They finally propose defense using a pre-trained model whose last layer is fine tuned. Several criticisms of novelty of the work in comparison to prior work were raised during the reviews. At the same time the validity and effectiveness of defenses remains unsubstantiated to a large extent. Although the authors put an effort in addressing these concerns, for the most part some reviewers and myself remain critical of the contributions and novelty of this work.